# Visceral Adipose Tissue Radiodensity Is Linked to Prognosis in Hepatocellular Carcinoma Patients Treated with Selective Internal Radiation Therapy

**DOI:** 10.3390/cancers12020356

**Published:** 2020-02-04

**Authors:** Maryam Ebadi, Carlos Moctezuma-Velazquez, Judith Meza-Junco, Vickie E. Baracos, Abha R. DunichandHoedl, Sunita Ghosh, Philippe Sarlieve, Richard J. Owen, Norman Kneteman, Aldo J. Montano-Loza

**Affiliations:** 1Division of Gastroenterology & Liver Unit, University of Alberta, Edmonton, AB T6G 2X8, Canada; ebadi@ualberta.ca (M.E.); moctezum@ualberta.ca (C.M.-V.); 2Department of Oncology, Cross Cancer Institute, Edmonton, AB T6G 1Z2, Canada; mezajunc@ualberta.ca (J.M.-J.); vbaracos@ualberta.ca (V.E.B.); abha@ualberta.ca (A.R.D.); sunita.ghosh@albertahealthservices.ca (S.G.); 3Department of Radiology and Diagnostic Imaging, Edmonton, AB T6G 2B7, Canada; psarlieve@yahoo.ca (P.S.); rowen@ualberta.ca (R.J.O.); 4Division of Transplantation, Department of Surgery, University of Alberta Hospital, Edmonton, AB T6G 2B7, Canada; kneteman@ualberta.ca

**Keywords:** radioembolization, mortality, adverse events, body composition, CT attenuation

## Abstract

Hepatocellular carcinoma (HCC) constitutes the fourth leading cause of cancer-related mortality. Various factors, such as tumor size, tumor multiplicity, and liver function, have been linked to the prognosis of HCC. The aim of this study was to explore the prognostic significance of muscle, subcutaneous and visceral adipose tissue (VAT) mass, and radiodensity, in a cohort of 101 HCC patients treated with selective internal radiation therapy (SIRT). Muscle and adipose tissue cross sectional area (cm^2^/m^2^) and radiodensity, reported as the Hounsfield Unit (HU), were determined using pre-SIRT computed tomography images. Cox proportional hazard models and exact logistic regression were conducted to assess associations between body composition and adverse outcomes. Majority of the patients were male (88%) with a mean VAT radiodensity of −85 ± 9 HU. VAT radiodensity was independently associated with mortality (HR 1.05; 95% CI: 1.01–1.08; *p* = 0.01), after adjusting for cirrhosis etiology, Barcelona Clinic Liver Cancer stage, previous HCC treatment, and portal hypertension markers. Patients with a high VAT radiodensity of ≥–85 HU had a two times higher risk of mortality (HR 2.01, 95% CI 1.14–3.54, *p* = 0.02), compared to their counterpart. Clinical features of portal hypertension were more prevalent in patients with high VAT radiodensity. High VAT radiodensity was associated with severe adverse events after adjusting for confounding factors. High VAT radiodensity is independently associated with both increased mortality and severe adverse events in patients treated with SIRT. VAT radiodensity measurement might serve as an objective approach to identify patients who will experience the most benefit from SIRT.

## 1. Introduction

Hepatocellular carcinoma (HCC) is the main form of primary liver cancer and constitutes the fourth leading cause of cancer-related mortality worldwide [1]. Regardless of its poor prognosis, locoregional therapies such as transarterial chemoembolization (TACE) and selective internal radiation therapy (SIRT) are available as treatment options for unresectable HCC [2]. SIRT with yttrium 90 (90Y-SIRT) is the most popular radioembolization technique [3] to reduce tumor burden, and improve the survival and quality of life of HCC patients [4]. 

Various factors, such as size and number of tumors, liver function, and body composition features have been linked to the prognosis of patients with HCC [2,5]. With regard to other predictive tests, the albumin–bilirubin (ALBI) grade demonstrates superiority to Child–Pugh in predicting the survival of HCC patients treated with radioembolization [6]. However, ALBI alone does not include other prognostic factors such as patient performance status and tumor burden. The Barcelona Clinic Liver Cancer (BCLC) staging system, which includes tumor features, liver function, and patient performance status, is an efficient tool to refine treatment options [7]; nevertheless, its ability to predict relevant clinical outcomes is questionable, particularly in the BCLC B stage, which is a very heterogeneous group. Among pre-treatment risk factors that result in poor prognosis, abnormal body composition features such as sarcopenia, and high visceral adiposity have recently attracted considerable attention [5], with a view to provide an objective assessment of the patient’s nutritional and metabolic status. 

Computed tomography (CT) image analysis has appeared as a rapid, precise, and accurate body composition assessment technique in order to quantify muscle and different adipose tissue depots [8]. Using this technique, sarcopenia has been linked to poor prognosis in patients with HCC [9], whereas high visceral adiposity was associated with higher probability of HCC incidence, recurrence, and worse outcomes [5,10]. However, a major gap remains regarding body composition association with outcomes in patients with HCC who undergo locoregional therapies. This led us to explore the prognostic significance of pre-SIRT body composition features, including muscle and adipose tissue depots, i.e., subcutaneous adipose tissue and visceral adipose tissue (VAT) mass and radiodensity, as potential factors to predict mortality in a retrospective cohort of 101 HCC consecutive patients treated with SIRT. In addition, we evaluated the association between pre-SIRT body composition features, treatment response, and severe adverse events (SAEs) post-SIRT.

## 2. Results

### 2.1. Baseline Characteristics of Patients 

Patient characteristics pre-SIRT are presented in Table 1. 

Eighty-nine patients were male (88%), with the mean age of patients 62 ± 12 years. The main cause of cirrhosis was hepatitis C (31%), followed by hepatitis B (21%), alcohol (14%), alcohol and hepatitis C (12%), and nonalcoholic steatohepatitis (6%). Other cirrhosis etiologies were hemochromatosis, autoimmune liver disease, and cryptogenic cirrhosis (17%). Most patients were in Child–Pugh class A, 69 patients (68%), with 32 patients in class B. Mean ALBI score was −2.3 ± 0.6. Patients belonged mainly to the BCLC stage B (46%), followed by A (29%) and C (25%). Among 38 patients with the history of previous treatment, TACE was the most common (45%), followed by combined treatment (24%), radiofrequency ablation (13%), surgical resection (10%), and percutaneous ethanol injection (8%).

Radiological evidence of vascular invasion was present in 19% of patients, of whom 37% had the involvement of the main portal vein and the remaining 63% experienced invasion of the right or left branches of the portal veins. Forty-two patients (42%) had a single tumor, and 58% had two or more tumors. The average largest dimension of tumors was 7 ± 4 cm. Complete response, partial response, and stable disease was achieved in 11%, 39%, and 29% of cases. Disease control rate was 78% and objective response rate was 50%. 

Mean BMI was 26 ± 4 kg/m^2^ and 56% of patients were sarcopenic. A summary of the body composition features is presented in Table 1. 

Patients were followed for a median time of 14 months (95% CI, 12–16) post-SIRT, until death (n = 65), or censoring (n = 36). Of 65 patients who died, 40 patients (62%) died from HCC progression and hepatic failure, and 25 patients due to other reasons, such as respiratory and renal failure. 

Overall mean VAT radiodensity was –85 ± 9 HU (range −106 to −64) and among body composition features, only VAT radiodensity (−83 ± 8 vs. −90±10 HU; *p* < 0.001) was higher in patients who died compared to the non-deceased patients.

### 2.2. Features Associated with Mortality in HCC Patients Post-SIRT

Through a univariate Cox proportional hazard analysis assessing the association between clinical and body composition features with mortality, hepatitis C and nonalcoholic-steatohepatitis-induced cirrhosis, radiological evidence of vascular invasion, ALBI score, Child–Pugh, BCLC stage, history of previous treatment, surrogate markers of portal hypertension (esophageal varices, splenomegaly with low platelet count), and VAT radiodensity were found to be predictors of mortality (Table 1). Considering the overlap between BCLC stages with vascular invasion, ALBI score and Child–Pugh, only the BCLC stage was included in multivariate analysis to avoid multicollinearity in the model. 

In multivariate analysis, VAT radiodensity, as a continuous variable, was significantly associated with mortality (HR 1.05, 95% CI, 1.01–1.08, *p* = 0.01; Table 1) after adjusting for the BCLC stage, previous treatment, markers of portal hypertension, and cirrhosis etiology. Every HU increase in VAT radiodensity was associated with 5% increased mortality risk.

The ability of the VAT radiodensity to rank patients according to mortality (i.e., C-statistic) was 0.73 (95% CI, 0.62–0.84, *p* < 0.001). Applying the highest Youden’s Index, VAT radiodensity of −85 (HU) was found to be independently associated with mortality. Patients with a high VAT radiodensity of ≥–85 (HU) had a two times higher risk of mortality (HR 2.01, 95% CI 1.14–3.54, *p* = 0.02; Table 2) compared to the patients with low VAT radiodensity (<−85 HU), after adjusting for the confounding factors.

Kaplan–Meier analysis of survival probability revealed that patients with a high VAT radiodensity survived for a median time of 10 months (95% CI, 9–11), while the median survival in the low radiodensity group was 21 months (95% CI, 8–33) (*p* < 0.001; Figure 1). 

Figure 2 highlights the VAT radiodensity quantification at L3 from two patients with similar visceral adipose tissue index (VATI). Figure 2a presents a patient with a VATI of 65 cm^2^/m^2^ and a low mean VAT radiodensity (−98 HU), whereas the patient in Figure 2b had a high mean VAT radiodensity (−80 HU) and a VATI of 64 cm^2^/m^2^. Increased mean VAT attenuation is displayed as an increase in the areas of high-radiodensity VAT (−50 to −85 HU). Areas demarked in yellow as low radiodensity (−86 to −150 HU) was predominant (77%) in Figure 2a, whereas 60% of the total VAT area in Figure 2b are areas composed of a high radiodensity VAT (−50 to −85 HU), tinted in pink.

### 2.3. Characteristics of Patients with High VAT Radiodensity

Patients with a high VAT radiodensity were younger, had a higher ALBI score, and a higher frequency of the Child–Pugh B class. No significant difference was observed between patients with low and high VAT radiodensity, with regards to sex, vascular invasion, previous treatment, or BCLC stage.

We observed a negative moderately strong linear correlation (r= −0.75, *p* < 0.001), between VATI and VAT radiodensity (Figure 3).

Regardless of having a higher VAT radiodensity, VATI (27 ± 16 vs. 64 ± 29 cm^2^/m^2^; *p* < 0.001) and SATI (44 ± 26 vs. 65 ± 37 cm^2^/m^2^; *p* = 0.002) were lower in patients with high VAT radiodensity, compared to those with a low VAT radiodensity. Muscle radiodensity was higher in patients with a high VAT radiodensity (36 ± 7 vs. 31 ± 8 HU; *p* < 0.001), whereas no difference was observed in SMI (Table 3).

Features of portal hypertension, including splenomegaly, ascites, and esophageal varices were more common in patients with a high VAT radiodensity. No significant difference in the presence of hepatic encephalopathy was observed between patients with high and low VAT radiodensity (Table 3). The frequency of liver failure-related mortality (73% vs. 50%, *p* = 0.07) tended to be higher in patients with a high VAT radiodensity.

### 2.4. Association of VAT Radiodensity with Response Assessment and Toxicities 

No association between VAT radiodensity and treatment response was found. Mean VAT radiodensity was −86 ± 9 and −86 ± 10 in patients with complete and partial response, respectively. No significance difference in disease control rate (75% vs. 80%, *p* = 0.61) and objective response rate (48% vs. 51%, *p* = 0.83) was observed between patients with high and low VAT radiodensity. Fourteen patients (14%) experienced SAEs after SIRT, of which ten developed liver decompensation, one acute cholecystitis, one patient developed a liver abscess, one needed to be hospitalized for post-embolization syndrome, and one had bleeding at the site of vascular access. Of those 14 patients with SAEs, 12 (86%) had high VAT radiodensity. Baseline features significantly associated with the development of post-SIRT SAEs in univariate analysis were Child–Pugh B, markers of portal hypertension and high VAT radiodensity (Table 4). High VAT radiodensity was independently associated with SAEs.

### 2.5. Performance High VAT Radiodensity to Predict SAEs after SIRT 

Presence of high VAT radiodensity had a sensitivity of 85.71% (95% CI 57.19–98.22), specificity of 59.77% (95% CI 48.71–70.15), a positive predictive value of 25.53% (95% CI 9.72–32.37), and a negative predictive value of 96.30% (95% CI 87.69–98.96) to predict the development of SAEs after SIRT. 

Comparison was made using the Child–Pugh score (A and B stages) to predict the development of SAEs after SIRT, where sensitivity was 64.29% (95% CI 35.14–87.24), specificity was 73.56% (95% CI 63.02–82.45), positive predictive value was 28.12% (95% CI 18.8–39.81), and the negative predictive value was 92.75% (95% CI 86.24–96.32).

## 3. Discussion

Association between baseline body composition features in HCC patients and post-SIRT outcomes was investigated in this study. Among body composition features, VAT radiodensity demonstrated appropriate capability to predict SAEs, after SIRT in HCC patients, as the probability (negative predictive value) was less than 5% in patients with a low VAT radiodensity, having a better performance than the Child–Pugh score. In addition, patients with a low VAT radiodensity had a better survival after SIRT. Therefore, this study presents a novel objective approach to estimate the risk of SAEs and mortality risk post-SIRT, given the baseline body composition features. This might serve as an approach to reserve SIRT for objectively selected patients.

VAT radiodensity represents a more objective predictor than the conventional risk factors for SAEs and survival after SIRT treatment. Moreover, association between VAT radiodensity but not the cross sectional area, with adverse outcomes in HCC patients treated with SIRT, suggests the prognostic significance of adipocyte remodeling with diminished lipid stores rather than the adipose tissue mass, in predicting the worse outcomes. This imaging technique offers valuable clinical insights into the prognosis of patients with HCC and outlines the necessity to conduct sensitive measures of VAT radiodensity for identifying mortality risk and severe adverse events in patients treated with SIRT, in order to target the particular populations that would benefit the most form SIRT. 

Our results are consistent with the Parikh et al. [12] study in a cohort of 75 patients with HCC who underwent TACE. They revealed that high VAT radiodensity was linked to shorter survival, portal hypertension, and development of hepatic decompensation [12]. High VAT radiodensity at the time of TACE in patients with a history of ascites was hypothesized to be an early marker of portal hypertension [12]. However, only ascites as a marker of portal hypertension was investigated in this study. The clinical practice guidelines from EASL and the European Organization for Research and Treatment of Cancer (EASL–EORTC) recommend the presence of esophagogastric varices, splenomegaly, or thrombocytopenia as surrogate markers of portal hypertension [13]. In our study, we found that patients with high VAT radiodensity had a higher rate of portal hypertension features, including splenomegaly, ascites, and esophageal varices. In line with the previous research [12], this suggests that VAT radiodensity might serve as an indirect marker of subclinical portal hypertension, which confers a poor prognosis in these patients. It is well known that hepatic vein pressure gradient (HVPG) in general, and clinically significant portal hypertension (i.e., HVPG > 10 mmHg) in particular, are the best predictors of clinical decompensation and poor survival in patients with cirrhosis [14]. However, because hepatic vein catheterization is an invasive procedure, not widely available outside of referral centers, it would be ideal to have a reliable surrogate marker. VAT radiodensity could be an indirect, accessible, and easy-to-measure maker of portal hypertension, and its association with HVPG should be explored. 

Radiodensity measured by CT HUs might be affected by various potential factors, such as blood flow, temperature [15], adipocyte size [16], and fluid-to-triglyceride ratio [17]. Therefore, HU does not solely measure stored triglyceride content in adipose tissue but also water, blood, and residual fat cell components [17]. In this study, high VAT radiodensity was defined as radiodensity above −85 (HU). The CT HU of brown adipose tissue (−10 to −87 HU) is significantly higher than that of white adipose tissue, which is about −88 to −190 HU [18]. Contrary to the white adipose tissue which is specialized to store energy in the form of triglycerides, brown/beige adipose tissue oxidizes fat and dissipates energy as heat. White adipose tissue browning has been recently identified as a contributor to energy wasting in cachexia [19]. Therefore, increased VAT radiodensity to the range of brown adipose tissue in patients with adverse outcomes might represent browning of the white adipose tissue, which contributes to increased lipid utilization and energy wasting. 

Additionally, elevated pressure in the portal vein might lead to the accumulation of protein-containing fluid within the abdominal cavity. It was postulated that VAT might have increased blood perfusion in patients with portal hypertension features, due to the diminished blood flow into liver. Increase in tissue-retained blood volume was associated with increase in brown adipose tissue radiodensity [17]. In agreement with this, the distribution of high VAT radiodensity seen in Figure 2 seems to be confined to the intraperitoneal areas drained by the portal system, as compared to the retroperitoneal areas, which bear a greater relation to the systemic circulation.

Contrary to previous studies showing an association between sarcopenia and poor prognosis in patients with HCC [9]; sarcopenia was not associated with mortality and SAEs in this group of patients with relatively preserved liver function. Sarcopenia has been linked to mortality in male patients with cirrhosis, whereas low subcutaneous adiposity is associated with higher mortality in female patients [20]. The metabolic pattern associated with fat loss in female patients with cirrhosis is comparable to chronic diseases or starvation, whereas the metabolic link to muscle loss in male patients is similar to the critical disease [21]. Moreover, sarcopenia was mainly related to mortality in patients with decompensated cirrhosis, whereas adipose tissue atrophy (adipopenia) was associated with mortality in a group of patients with compensated cirrhosis [22]. Although around 90% of the patients in this study were male, they might not yet have been placed in the catabolic state associated with the depletion of muscle tissue and were not experiencing the severe exhaustion of body muscle reservoirs that happens at later stages of the disease. Additionally, previous studies found that white adipose tissue browning occurs in the initial stages of cachexia, prior to skeletal muscle atrophy [19]. Therefore, various factors such as the stage of cirrhosis and presence of decompensation might influence the prognostic value of components of body composition.

We acknowledge there are limitations in this study, as we were unable to evaluate other mortality-associated factors such as the tumor dose deposition in patients, due to the retrospective nature of the study. Considering the enrollment time and the impact of learning curves on improved treatment delivery, the long inclusion time, and subsequently, the changes in SIRT procedure might be another limitation of this study. In addition, considering the limited number of patients in this study and consequently the small number of SAEs, there is not enough power to fully understand the correlation between VAT radiodensity with SAEs. Additionally, we were not able to effectively grade clinical SAEs, given the nature of the study. Lastly, some patients were not qualified for inclusion as no CT scans could be retrieved for these patients. We recognize that cut-offs established in this study might not be generalized to other patient populations and, therefore, they should be validated in an external cohort of patients with HCC treated with SIRT.

## 4. Materials and Methods 

### 4.1. Study Population

The study was reviewed and approved by the Institutional Review Board of the University of Alberta (Pro00066572). All adult patients (≥18 years) who underwent SIRT as the primary treatment for a non-resectable HCC at a single center between October 2006 and June 2015 were enrolled in the study (n = 126). All patients were receiving SIRT for the first time when enrolled in the study. 

Diagnosis of HCC was based on the European Association for the Study of the Liver (EASL) practice guidelines [23]. The 2017 version of the Liver Imaging Reporting and Data System (Li-RADS) approved by the American College of Radiology was used to analyze the radiological patterns and homogenize the explanation of multiple CT and MRI images for HCC. Patients without appropriate CT images at the time of the first SIRT session (n = 15), as well as patients who underwent liver transplantation (n = 10) were excluded from the study. Clinical and demographic features of the patients were collected from medical charts.

The decision to select patients for SIRT was made in the multidisciplinary rounds of the University of Alberta hospital, composed of interventional radiologists, hepatologists, hepatobiliary surgeons, and oncologists. We considered patients for SIRT if they were not candidates for surgical resection or ablation (i.e., stage migration strategy) and at the same time were not suitable for TACE, due to portal vein thrombosis or portal vein invasion, multifocal tumors or larger tumors. This was a rescue therapy for patients who failed TACE with progressive disease by mRECIST/EASL after two attempts, or liver transplant candidates who qualified for down staging [24].

### 4.2. SIRT Procedure, Toxicities, and Response Assessment

SIRT using yttrium-90 glass microsphere dosage was based on a simplified partition model. Hepatic angiography was carried out on all patients, and in order to assess the pulmonary shunt fraction, confirm activity within the tumor vasculature, and eliminate non-target delivery of therapy in the determined catheter position, a celiac-mesenteric angioscintigraphy with 150 MBq technetium-99 m labeled macroaggregated albumin (99mTc-MAA) scanning was implemented. Partition model with dose ranges of 100–120 Gy (1 Gy = 1 Joule /Kg of tissue) was used to estimate treatment doses that were performed two to four weeks post angioscintogram, using Y90-glass microspheres (TheraSphere©, BTG International Canada Inc., Ottawa, ON, Canada). Two experienced interventional radiologists (P.S.; R.J.O.) performed the angiography, dose calculations, and imaging evaluations.

A follow-up triphasic CT scan or contrast enhanced MRI was attained 10–12 weeks post-SIRT, and subsequently every 12 weeks, to evaluate response. Modified Response Evaluation Criteria in Solid Tumors (mRECIST) was applied to determine the radiological response [25]. The objective response rate was the proportion of patients that attained the best response of complete or partial response, and the disease control rate was the percentage of patients that reached a best response of complete response, partial response, or stable disease.

Development of severe adverse events was evaluated one-to-two weeks post-SIRT, and every three months, thereafter. Laboratory toxicities were graded according to the Common Terminology Criteria for Adverse Events v4.0; grade 3 or higher toxicities were categorized as SAEs. For clinical toxicity, any clinical event requiring hospital admission or attendance was considered to be SAE, due to the retrospective nature of the study.

### 4.3. CT Image Analysis

Abdominal CT scans taken at the third lumbar vertebra (L3) were analyzed using the Slice-O-Matic software (V4.2; Tomovision, Montreal, QC, Canada), as a part of the pre-SIRT workup. Skeletal muscle and adipose tissue areas estimated from a single CT image at L3, was found to best correlate with the whole body muscle (R^2^ = 0.855, *p* < 0.01) and adipose tissue mass (R^2^ = 0.927, *p* < 0.001); this has been used as the appropriate anatomical landmark [8]. There is also a strong correlation between VAT cross-sectional areas at L3 and the whole-body VAT volume [26]. Cross-sectional area for each tissue was quantified using standard Hounsfield Unit (HU) thresholds of −29 to 150 HU for skeletal muscle, −150 to −50 HU for VAT [27] (adipose tissue inside the abdominal wall), and −190 to −30 HU for subcutaneous adipose tissue (area contained between the skin and the outer surface of the abdominal wall muscles) [28]. Cross sectional areas of these tissues were quantified by summing up tissue pixels and multiplying by the pixel surface areas. Given the effect of contrast medium administration on muscle and VAT cross-sectional area and radiodensity estimation [29], only CT slices without contrast were selected for body composition analysis in this study. Using non-contrast abdominal CT images, high intraobserver and interobserver reproducibility was observed in measuring the body composition parameters in clinical populations [29]. Using software such as Slice-O-Matic, tissues were selected semi-automatically and the average time to quantify body composition parameters was found to be around eight minutes, which can be easily carried out by any trained observers with a knowledge of human anatomy [30]. 

Three body composition variables; visceral adipose tissue index (VATI), subcutaneous adipose tissue index (SATI), and skeletal muscle index (SMI) were determined by normalizing the tissue areas (cm^2^) to the squared patient height (cm^2^/m^2^). Cut-off values to define sarcopenia in HCC patients treated with SIRT has not been standardized and, therefore, sarcopenia in this study was defined using pre-established cut-offs in patients with cirrhosis awaiting liver transplantation, as SMI is <39 cm^2^/m^2^ in females and <50 cm^2^/m^2^ in males [11]. For radiodensity assessment, mean tissue attenuation (HU) was reported for the entire cross-sectional area at L3, separately for muscle, SAT, and VAT.

### 4.4. Statistical Analysis

Continuous variables were reported as mean and standard deviation (SD) and independent *t*-test was applied to compare the differences in means. For categorical variables, descriptive statistics were presented as percentages and Fisher’s exact test was used to determine associations between categorical variables. Correlation between VATI and VAT radiodensity was determined by Pearson’s correlation coefficient (*r*) analysis.

Overall survival was the main outcome of the study defined as the time from the first SIRT to the date of death or date of last visit. In order to determine significant predictors of mortality, univariate and multivariate Cox proportional hazard models were created to determine mortality hazard ratios (HRs), along with their 95% confidence intervals (CI). The multivariate model included variables with *p* < 0.10 in the univariate analysis. The most efficient model was designated as the concluding model. To avoid the consequence of collinearity, among ALBI grade, Child–Pugh, and BCLC stage, only BCLC stage was included in the multivariate analysis.

First, we assessed the capability of body composition features to predict mortality, as dimensional variables. Subsequently, optimal cut-off value to predict mortality was established only for the significant dimensional body composition variables, using a receiver-operating characteristic analysis. Proficiency of the model to discriminate between outcome clusters was evaluated using the area under the curve. Values with the maximum Youden’s Index (sensitivity + specificity − 1) were incorporated into adjusted multivariate models and the value with the highest significant *p*-value was deliberated as the optimal cut-off.

Overall survival over time was estimated by plotting the Kaplan–Meier curves and comparisons between low and high VAT radiodensity survival curves was conducted using the log-rank test. Exact logistic regression, rather than the binary logistic regression, was conducted to determine factors associated with higher risk of SAEs. This method is useful when the sample size is too small for a regular logistic regression. We had only 14 patients with SAEs, and therefore, exact logistic regression was conducted. However, it should be mentioned that this analysis preferentially works best with dichotomous covariates and, therefore, patients in BCLC stages B and C were merged together for this analysis [31].

Sensitivity, specificity, positive predictive value, and negative predictive value for SAEs were calculated, according to the frequency of high and low VAT radiodensity. 

## 5. Conclusions

In summary, post-SIRT survival is influenced by various factors related to liver function and tumor burden. However, this study demonstrated the prognostic value of VAT radiodensity in patients with HCC undergoing SIRT. Association between high VAT radiodensity and adverse outcomes in patients with cirrhosis might represent remodeling of adipose tissue with diminished lipid stores. Given the survival discrimination of high VAT radiodensity in patients with HCC, VAT radiodensity will be beneficial for clinicians to determine whether SIRT should be recommended to patients with HCC and to identify patients who will experience the most survival advantage of the treatment and have a low risk of SAE development.

## Figures and Tables

**Figure 1 cancers-12-00356-f001:**
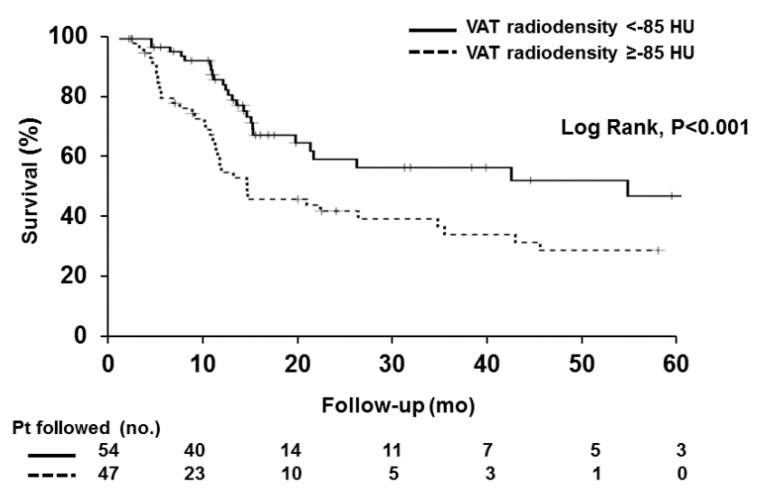
Survival curves in patients with high versus low visceral adipose tissue radiodensity. Survival over time was assessed using Kaplan–Meier curves and the curves were compared using the log-rank test. Shorter median survival was observed in patients with high visceral adipose tissue (VAT) radiodensity, compared to the patients with low VAT radiodensity (Log rank < 0.001).

**Figure 2 cancers-12-00356-f002:**
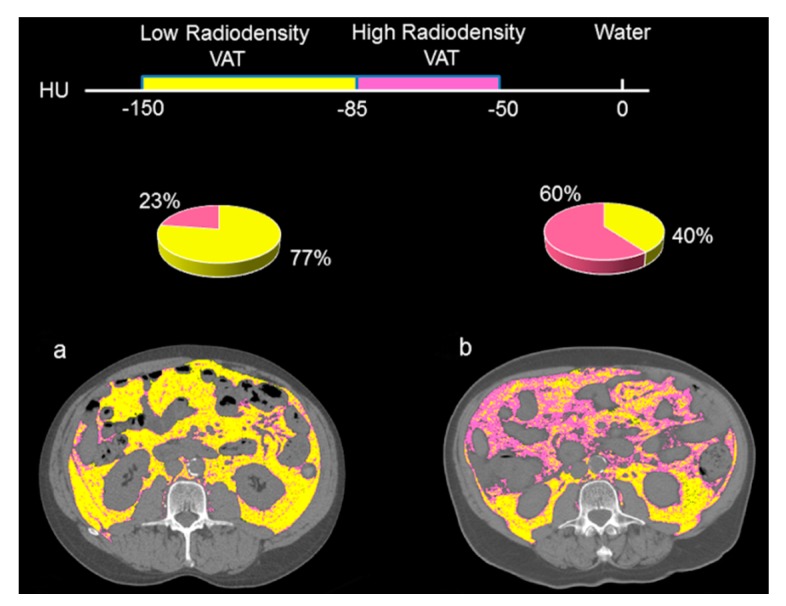
Abdominal CT images taken at the 3rd. lumbar vertebra of patients with high versus low visceral adipose tissue radiodensity. Comparison of two HCC patients with the same visceral adipose tissue index (**a**) with a low visceral adipose tissue (VAT) radiodensity (−98 HU) and (**b**) with a high VAT radiodensity (−80 HU). Visceral adipose tissue with high radiodensity (−50 to −85) is shown in pink and low radiodensity VAT (−86 to −150) is shown in yellow. More than 75% of the total VAT area in Figure 2a represents the area composed of low-radiodensity VAT, whereas for Figure 2b the areas of high-radiodensity VAT is predominant (60%).

**Figure 3 cancers-12-00356-f003:**
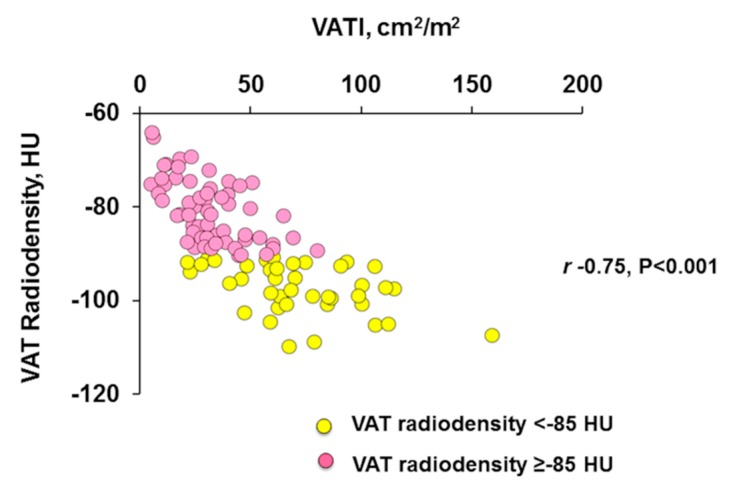
Scatter graph depicting correlations between visceral adipose tissue index and radiodensity. Negative moderately strong correlation (Pearson’s correlation) between visceral adipose tissue index (VATI) and VAT radiodensity in patients with HCC (r= −0.75, *p* < 0.001).

**Table 1 cancers-12-00356-t001:** Pre-selective internal radiation therapy (SIRT) factors associated with mortality, as per the Cox proportional-hazards analysis.

				Univariate		Multivariate	
Characteristics	All patients (n = 101)	Censored (n = 36)	Dead (n = 65)	HR (95% CI)	*p*-value	HR (95% CI)	*p*-value
Sex, male	89 (88)	32 (89)	57 (88)	0.92 (0.44–1.94)	0.83		
Age at SIRT, years	62 ± 12	61 ± 19	62 ± 11	0.996 (0.98–1.01)	0.66		
Cirrhosis etiology							
- Alcohol	14 (14)	5 (14)	9 (14)	1.04 (0.51–2.10)	0.92		
- HCV	31 (31)	8 (22)	23 (35)	2.43 (1.43–4.13)	0.001	2.28 (1.28–4.04)	0.005
- Alcohol and HCV	12 (12)	3 (8)	9 (14)	1.92 (0.86–4.27)	0.11		
- HBV	21 (21)	10 (28)	11 (17)	0.69 (0.35–1.35)	0.28		
- NASH	6 (6)	4 (11)	2 (3)	0.23 (0.05–0.94)	0.04	0.88 (0.19–4.16)	0.87
- Others ^a^	17 (17)	6 (17)	11 (17)	0.62 (0.32–1.18)	0.15		
Vascular invasion ^b^	19 (19)	5 (14)	14 (22)	1.78 (0.96–3.32)	0.07		
Extrahepatic spread	4 (4)	0 (0)	4 (6)	1.12 (0.41–3.12)	0.82		
ALBI score ^b^	−2.3 ± 0.6	−2.5 ± 0.5	-2.2±0.6	2.82 (1.78–4.47)	<0.001		
Child–Pugh ^b^							
- A	69 (68)	31 (86)	38 (59)				
- B	32 (32)	5 (14)	27 (42)	2.79 (1.68–4.63)	<0.001		
Number of tumors ^b^	4 ± 5	3 ± 4	5±6	1.1 (1.02–1.19)	0.02		
Largest dimension of tumors	7 ± 4	8 ± 5	7 ± 4	0.97 (0.87–1.09)	0.62		
BCLC stage							
- A	29 (29)	17 (47)	12 (19)				
- B	47 (46)	14 (39)	33 (51)	2.74 (1.40–5.34)	0.003	2.80 (1.37–5.73)	0.005
- C	25 (25)	5 (14)	20 (31)	2.93 (1.42–6.06)	0.004	2.40 (1.07–5.40)	0.04
Previous treatment	38 (38)	8 (22)	30 (46)	2.33 (1.41–3.85)	0.001	1.58 (0.90–2.76)	0.11
Markers of portal hypertension ^c^	60 (60)	19 (53)	41 (63)	2.07 (1.24–3.45)	0.005	1.38 (0.80–2.37)	0.25
Body Composition							
BMI, kg/m^2^	26 ± 4	25 ± 4	26 ± 6	1.04 (0.97–1.11)	0.29		
SMI (cm^2^/m^2^)	47 ± 10	47 ± 9	48 ± 10	1.01 (0.98–1.03)	0.64		
VATI (cm^2^/m^2^)	47 ± 30	53 ± 35	43 ± 27	0.996 (0.99–1.004)	0.32		
SATI (cm^2^/m^2^)	55 ± 34	61 ± 43	51 ± 27	1.002 (0.99–1.01)	0.73		
Muscle radiodensity, (HU)	33 ± 8	32 ± 8	34 ± 8	1.00 (0.97–1.03)	0.96		
Visceral adipose radiodensity, (HU)	−85 ± 9	−90 ± 10	−83 ± 8	1.06 (1.03–1.09)	<0.001	1.05 (1.01–1.08)	0.01
Subcutaneous adipose radiodensity, (HU)	−93 ± 12	−96 ± 13	−92 ± 11	1.02 (0.996–1.04)	0.12		
Sarcopenia ^d^	57 (56)	22 (61)	35 (54)	1.44 (0.75–2.79)	0.28		

ALBI, albumin-bilirubin score; BCLC, Barcelona-Clinic Liver Cancer; CI, confidence interval, HBV, hepatitis B; HCV, hepatitis C; HR, hazard Ratio; HU, Hounsfield unit; NASH, non-alcoholic steatohepatitis; SATI, subcutaneous adipose index; SIRT, selective internal radiation therapy; SMI, skeletal muscle index; VATI, visceral adipose tissue index. ^a^ Other causes of cirrhosis were Cryptogenic (n = 13), Hemochromatosis (n = 3), and Autoimmune liver disease (n = 1). ^b^ Were not included in multivariate analysis to avoid collinearity with BCLC stage. ^c^ Markers of portal hypertension includes splenomegaly, esophageal varices or thrombocytopenia (platelet count <100,000/mm^3^). ^d^ Sarcopenia was defined using established cut-offs in patients with cirrhosis [11].. Numbers in parentheses are percentages.

**Table 2 cancers-12-00356-t002:** Features associated with mortality through a univariate and multivariate Cox proportional hazards analysis.

Characteristic	Univariate Analysis		Multivariate Analysis	
	HR (95% CI)	*p*-value	HR (95% CI)	*p*-value
HCV-induced HCC	2.43 (1.43–4.13)	0.001	2.17 (1.21–3.87)	0.009
NASH-induced HCC	0.23 (0.05–0.94)	0.04	0.74 (0.16–3.43)	0.70
BCLC stage				
- A				
- B	2.74 (1.40–5.34)	0.003	2.82 (1.37–5.81)	0.005
- C	2.93 (1.42–6.06)	0.004	2.72 (1.22–6.05)	0.01
Previous treatment	2.33 (1.41–3.85)	0.001	1.66 (0.96–2.86)	0.07
Markers of portal hypertension ^a^	2.07 (1.24–3.45)	0.005	1.37 (0.79–2.38)	0.26
High visceral adipose Radiodensity (VAT HU≥ −85 HU)	2.56 (1.54–4.26)	0.002	2.01 (1.14–3.54)	0.02

BCLC, Barcelona-Clinic Liver Cancer; CI, confidence interval, HCV, hepatitis C; HR, hazard Ratio; NASH, non-alcoholic steatohepatitis. ^a^ Markers of portal hypertension includes splenomegaly, esophageal varices or thrombocytopenia (platelet count <100,000/mm^3^). HRs and P values were estimated using Cox proportional hazard model.

**Table 3 cancers-12-00356-t003:** Clinical features associated with high VAT radiodensity.

Characteristics	High VAT radiodensity (n = 47)	Low VAT radiodensity (n = 54)	*p*-value
Sex, male	39 (83)	50 (93)	0.22
Age at SIRT, years	60 ± 14	65 ± 11	0.046
Cirrhosis etiology			
- Alcohol	6 (13)	8 (15)	1.00
- HCV	17 (36)	14 (26)	0.29
- Alcohol and HCV	7 (15)	5 (9)	1.00
- HBV	9 (19)	12 (22)	0.81
- NASH	0 (0)	6 (11)	0.03
- Others	8 (17)	9 (17)	1.00
Vascular invasion	10 (21)	9 (17)	0.62
Extrahepatic spread	3 (6)	1 (2)	0.34
ALBI score	−2.1 ± 0.7	−2.5 ± 0.5	<0.001
Child–Pugh			
- A	25 (53)	44 (82)	0.003
- B	22 (47)	10 (18)	
BCLC stage			
- A	12 (26)	17 (32)	0.13
- B	19 (40)	28 (52)	
- C	16 (34)	9 (17)	
Previous treatment	22 (47)	16 (30)	0.10
Type of previous treatment			
- Surgical Resection	3 (14)	1 (6)	0.62
- PEI	1 (4)	2 (13)	0.57
- TACE	8 (37)	9 (56)	0.34
- RFA	3 (14)	2 (13)	1.00
- Combined Treatment	7 (32)	2 (13)	0.26
Splenomegaly	24 (51)	8 (15)	<0.001
Ascites	19 (40)	6 (11)	0.001
Esophageal varices	20 (43)	13 (24)	0.06
Encephalopathy	2 (4)	1 (2)	0.60
Thrombocytopenia	19 (40)	14 (26)	0.14
Body Composition			
BMI	24 ± 4	26 ± 4	0.02
SMI (cm^2^/m^2^)	47 ± 9	48 ± 10	0.35
VATI (cm^2^/m^2^)	27 ± 16	64 ± 29	<0.001
SATI (cm^2^/m^2^)	44 ± 26	65 ± 37	0.002
Muscle radiodensity, (HU)	36 ± 7	31 ± 8	<0.001
Visceral Adipose Radiodensity, (HU)	−77 ± 5	−93 ± 5	<0.001
Subcutaneous Adipose Radiodensity, (HU)	−87 ± 13	−99 ± 6	<0.001
Sarcopenia ^a^	27 (57)	30 (56)	1.00

ALBI, albumin-bilirubin score; BCLC, Barcelona-Clinic Liver Cancer; HBV, hepatitis B; HCV, hepatitis C; HU, Hounsfield unit; NASH, non-alcoholic steatohepatitis; PEI, percutaneous ethanol injection; RFA, radiofrequency ablation; SATI, subcutaneous adipose index; SIRT, selective internal radiation therapy; SMI, skeletal muscle index; TACE, transarterial chemoembolization; VATI, visceral adipose tissue index. ^a^ Sarcopenia was defined using established cut-offs in patients with cirrhosis [11]. Numbers in parentheses are percentages.

**Table 4 cancers-12-00356-t004:** Baseline features related to post-SIRT severe adverse events.

Characteristics	Univariate		Multivariate	
	OR (95% CI)	*p*-value	OR (95% CI)	*p*-value
Sex, male	0.78 (0.14–8.19)	1.00		
Child–Pugh B	4.92 (1.32–20.74)	0.01	2.48 (0.57–11.84)	0.28
BCLC stage	1.56 (0.37–9.40)	0.77		
Previous treatment	2.51 (0.69–9.66)	0.19		
Markers of portal hypertension ^a^	4.81 (0.98–46.82)	0.05	2.34 (0.37–25.68)	0.53
High visceral adipose radiodensity ^b^	8.74 (1.78–85.14)	0.003	5.61 (1.05–56.89)	0.04

^a^ Markers of portal hypertension includes splenomegaly, esophageal varices, or thrombocytopenia (platelet count <100,000/mm^3^). ^b^ Defined as visceral adipose radiodensity ≥ −85 HU. ORs and P values were estimated using Exact Logistic Regression. For the BCLC stage, patients in category A were compared to patients in categories B and C.

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
