# Peer review of "Visceral Adipose Tissue Radiodensity Is Linked to Prognosis in Hepatocellular Carcinoma Patients Treated with Selective Internal Radiation Therapy"

_cancers, 2020, doi:10.3390/cancers12020356_

Round 1
Reviewer 1 Report
In the manuscript "VAT radiodensity is linked to prognosis in HCC patients treated with SIRT", the authors investigate the association of anatomical features from pre-treatment CT with survival and treatment complications by analysing a retrospective cohort of 101 patients. They conclude that the radiodensity of visceral adipose tissue is a strong predictor of both survival and severe adverse events.
The manuscript is original, well written and interesting. The discussion contains speculative but interesting ideas.
The data has limitations, as recognised by the authors, and the data analysis is quite well set up to deal with these limitations and get the most out of the data.
The analyses methods used by the authors are a mixture of, on the one side, accurately estimating the effect size of VAT radiodensity on outcome, and on the other side, developing a prediction model. However, they fail to do both correctly.
The analysis may be improved by the following:
For accurately estimating the effect size of VAT radiodensity, the authors should correct for confounding factors instead of variables that are selected based on significance in univariable analysis. Furthermore, they should adjust for multiple testing because they select VAT radiodensity based on the data from a set of candidate predictors.
For prediction modeling, the authors should add internal validation (e.g. using bootstrapping or cross validation) to correct for optimism. Furthermore, they must specify the full model with formula and all coefficient, including intercepts or baseline hazard functions.
Specifying the models completely will be vital for others to validate the results in different cohorts. Also, this enables individual prediction for patients by others. A full model specification must be included in the manuscript, possibly as supplementary material.
Detail
In table 1 under "Previous treatment" there are five categories ('Surgical Resection' to 'Combined Treatment'), but there are 6 numbers of patients (38 to 9), with different definitions of the percentages between brackets, and there is only one HR. I think that this is meant as 'previous treatment vs no previous treatment', with additional specification of the subgroups of type of treatment in the 'previous treatment' group. This should be specified more clearly in the table, e.g., by splitting the types of information into two separate rows. A similar comment applies to Table 3.
Author Response
Thank you for your comment. In order to improve the analysis, our manuscript was revised and modified under the supervision of Dr. Sunita Ghosh (Biostatistician, Associate Clinical Professor, Department of Medical Oncology, University of Alberta, Cross Cancer Institute, Edmonton, AB, Canada).
As it was mentioned by previous researchers, confounding can be controlled for at the analytical stage through multivariate analysis (1. Aschengrau, A and Seage, G.R, 2009, Essential of epidemiology in public health. Sudbury, MA: Jones and Bartlett; 2. Kahlert J, et al. Control of confounding in the analysis phase - an overview for clinicians. Clin Epidemiol. 2017 Mar 31;9:195-204).
The aim of this study was not to construct a prediction model, but rather to evaluate the association of riskfactors with clinical outcome, in which associations are adjusted for confounders.
In order to prevent over-fitting or production of an analysis that corresponds too closely or exactly to our data set, we used the rule of the thumb of having at least 10 outcomes for each variable that we included in the models for VAT radiodensity and mortality (F.E. Harrell Jr.Regression modeling strategies. Springer-Verlag, New York, NY (2001). Given 65 death in this study, generous inclusion of potential confounding factors in the multivariable model is likely to be a problem; however, as requested by the reviewer, we repeated multivariate analysis including age and sex and noticed that high VAT radiodensity remained significant in the model (HR 2.02, 95% CI 1.13-3.60, P=0.02).
|
|
Univariate Analysis |
Multivariate Analysis |
||
|
Characteristic |
HR (95% CI) |
P-value |
HR (95% CI) |
P-value |
|
Sex, male |
0.92 (0.44-1.94) |
0.83 |
0.81 (0.36-1.84) |
0.62 |
|
Age at SIRT, years |
0.996 (0.98-1.01) |
0.66 |
1.01 (0.99-1.03) |
0.81 |
|
HCV- induced HCC |
2.43 (1.43-4.13) |
0.001 |
2.11 (1.17-3.79) |
0.01 |
|
NASH- induced HCC |
0.23 (0.05-0.94) |
0.04 |
0.71 (0.15-3.33) |
0.71 |
|
BCLC stage A B C |
2.74 (1.40-5.34) 2.93 (1.42-6.06) |
0.003 0.004 |
3.11 (1.45-6.65) 3.29 (1.38-7.85) |
0.004 0.007 |
|
Previous treatment |
2.33 (1.41-3.85) |
0.001 |
1.67 (0.96-2.91) |
0.07 |
|
Markers of portal hypertension |
2.07 (1.24-3.45) |
0.005 |
1.34 (0.76-2.36) |
0.02 |
|
High visceral adipose radiodensity |
2.56 (1.54-4.26) |
0.002 |
2.02 (1.13-3.59) |
0.02 |
For SAEs, We first included gender as a cofounding variable and noticed that high VAT radiodensity was still independently associated with SAEs.
|
|
Univariate |
Multivariate |
||
|
Characteristics |
OR (95% CI) |
P-value |
OR (95% CI) |
P-value |
|
Sex, male |
0.78 (0.14-8.19) |
1.00 |
1.20 (0.16-14.79) |
1.00 |
|
Child-Pugh B |
4.92 (1.32-20.74)
|
0.01
|
2.51 (0.57-12.15)
|
0.28
|
|
BCLC stage
|
1.56 (0.37-9.40)
|
0.77
|
|
|
|
Previous treatment |
2.51 (0.69-9.66) |
0.19 |
|
|
|
Markers of portal hypertension |
4.81 (0.98-46.82) |
0.05 |
2.23 (0.36-24.50) |
0.57 |
|
High visceral adipose radiodensity |
8.74 (1.78-85.14) |
0.003 |
5.71 (1.04-58.74) |
0.04 |
However, we were not able to include both age and gender as confounding factors as the model was unstable due to the limited events and too many variables.
Furthermore, they should adjust for multiple testing because they select VAT radiodensity based on the data from a set of candidate predictors.
Multiple testing would be applicable if we were testing multiple hypothesis or if we were comparing multiple groups. The purpose of our study is to determine the risk factors associated with the overall survival. Hence we used a standard model building approach where we selected variables associated with the outcome in an univariate setting and including the significant variables at the univariate to be included in the multivariate model. The approach we have followed to select the variables was based on the significance of adjusted HRs but not the strength of significance; therefore, we did not adjust definition of significance for multiple testing.
For prediction modeling, the authors should add internal validation (e.g. using bootstrapping or cross validation) to correct for optimism. Furthermore, they must specify the full model with formula and all coefficient, including intercepts or baseline hazard functions. Specifying the models completely will be vital for others to validate the results in different cohorts. Also, this enables individual prediction for patients by others. A full model specification must be included in the manuscript, possibly as supplementary material.
Thank you for your comment. For internal validation, we performed bootstrapping (500 number of samples) and results were in the same line.
Bootstap for variables in the Equation
|
|
B |
Bias |
Std. Error |
Sig. (2-tailed) |
BCa 95% Confidence Interval |
|
|
Lower |
Upper |
|||||
|
HCV- induced HCC |
.773 |
-.025b |
.358b |
.022b |
.118b |
1.389b |
|
NASH- induced HCC |
-.304 |
-3.000b |
5.148b |
.486b |
-12.193b |
.324b |
|
BCLC stage - A - B - C |
1.037 1.001 |
.034b .093b |
.456b .501b |
.018b .022b |
.107b .033b |
2.080b 2.451b |
|
Previous treatment |
.504 |
.050b |
.318b |
.076b |
-.167b |
1.391b |
|
Markers of portal hypertensiona |
.315 |
.108b |
.358b |
.324b |
-.466b |
1.498b |
|
High visceral adipose radiodensity (VAT HU≥ -85HU) |
-.696 |
-.024b |
.385b |
.034b |
-1.437b |
-.053b |
|
a. Unless otherwise noted, bootstrap results are based on 500 bootstrap samples |
|
b. Based on 499 samples |
The Cox model has no intercept (Cleves M, et al. An Introduction to Survival Analysis Using Stata, Second Edition 2008). For the SEAs, as we mentioned previously in the manuscritp “considering the limited number of patients in this study and consequently small number of SAEs, there is not enough power to fully understand the correlation between VAT radiodensity with SAEs.” (Lines 334-336), and we also mentioned that “cut-offs established in this study may not be generalized to other patients populations and therefore they should be validated in external cohort of patients with HCC treated with SIRT.” Given that the aim of this study was to explore the association but not creating a prediction model, no full model specification is presented.
Detail
In table 1 under "Previous treatment" there are five categories ('Surgical Resection' to 'Combined Treatment'), but there are 6 numbers of patients (38 to 9), with different definitions of the percentages between brackets, and there is only one HR. I think that this is meant as 'previous treatment vs no previous treatment', with additional specification of the subgroups of type of treatment in the 'previous treatment' group. This should be specified more clearly in the table, e.g., by splitting the types of information into two separate rows. A similar comment applies to Table 3.
Thank you for your suggestion, now we have modified Tables 1 and 3.
Thirthy-eight was the total number of patients with previous treatments and the remaining numbers were the number of patients in each category which sums up to 38. To make it clear, we removed the subgroups from the table and explained them in the text as follows:
Among 38 patients with the history of previous treatment, TACE was the most common (45%), followed by combined treatment (24%), radiofrequency ablation (13%), surgical resection (10%) and percutaneous ethanol injection (8%) (Lines 112-114).Table 3 was modified accordingly.
Reviewer 2 Report
Comment to the authors
In this paper Ebadi et al. presented retrospective data on the association between visceral adipose tissue radiodensity and survival data of cirrhotic patients affected by HCC and treated by SIRT. The topic is interesting since a better stratification and subsequent treatment allocation of BCLC B and C is urgently needed. The data from Ebadi et al show that high VAT radiodensity is associated with poor long term prognosis probably because it correlates with liver function and adverse events. However, the analysis should be refined and some aspects on patients selection should be clarified. Indeed, 58% of treated patients were BCLC A and SIRT is not a primary choice for these patients and the patients selection is not well described in the methods section. Patient follow-up should be extend if possible (1/3 of censored). Tumor response should be included in the multivariate analysis. VAT radiodensity seems to be correlated more to liver function than being specific for SIRT and cancer. Indeed, in the high VAT radiodensity group more patients were Child C (34% vs 17%) and ascites (40 vs 11%). Also, VAT HU levels have been correlated with overall survival in the Framingham cohort and survival after TACE.
Major Comments
-Methods section should be placed before results.
- Given the enrollment period and the median survival, survival data should be available almost for all the patients.
- It is not clear why these patients had SIRT and not alternative treatments. Indeed, SIRT is considered an alternative to Sorafenib in BCLC C patients or BCLC B patients non responding to multiple TACE.
- Statical analysis, continous variables were compared by t-test but normality was not assessed.
- Table 1, given the small sample size, data of the 2 groups (alive, dead) can be shown.
- Table 1, tumor number and size should be shown.
- Table 1 and 2, treatment response and SAE should be included.
- Table 1 and 2, since VAT radiodensity correlates with Child score and not with BCLC, it would be important to see how the its predictive value changes if Child score and vascular invasion and included in the model instead of BCLC.
- Definition of vascular invasion should be provided (segmental ? Main branch ?).
- Table 2, why the univariate analysis values for HCV changed from Table 2 ?
- Figure 1, censored should be shown
- Table 3, was the ascites present at the time of the SIRT or during the follow-up ?
- Discussion, the long enrollment time (7 years) should be discussed as a limitation since could affect the way how SIRT or VAT radiodensity were performed or assessed.
Minor comments
Table 2, specify in the characteristic the cut off for high VAT (not in the legend). Lines 427- 442, paragraph difficult to read.
Author Response
Major Comments
- Methods section should be placed before results.
Thank you for your comment; however, we followed the instructions for authors and therefore methods section is placed after results according to the Research Manuscript Sections.
- Given the enrollment period and the median survival, survival data should be available almost for all the patients.
We thank the reviewer for the comment; however, we were not able to extend the last follow-up date as majority of censored patients were out of province patients and therefore lost to follow-up and were censored at the time of last known contact.
- It is not clear why these patients had SIRT and not alternative treatments. Indeed, SIRT is considered an alternative to Sorafenib in BCLC C patients or BCLC B patients non responding to multiple TACE.
The decision to select patients for SIRT was made in the multidisciplinary rounds of the University of Alberta hospital, composed by interventional radiologists, hepatologists, hepatobiliary surgeons and oncologists. We consider patients for SIRT if they are not candidates for surgical resection or ablation (i.e. stage migration strategy), and at the same time are not suitable for TACE due to portal vein thrombosis or portal vein invasion, multifocal tumors or larger tumors, as a rescue therapy in patients who failed TACE with progressive disease by mRECIST/EASL after two attempts, or liver transplant candidates qualifying for down staging (Cancer Treat Rev. 2012 Feb;38(1):54-62. doi: 10.1016/j.ctrv.2011.05.002. Epub 2011 Jul 2. Locoregional radiological treatment for hepatocellular carcinoma; Which, when and how? Meza-Junco J1, Montano-Loza AJ, Liu DM, Sawyer MB, Bain VG, Ma M, Owen R).
We have included this in lines 354-360.
- Statical analysis, continous variables were compared by t-test but normality was not assessed.
Thank you for your comment. We consider that as this study includes 101 patients, by law of central limit theorem (sample size, n ≥30) it is assumed to be a normal distribution.
- Table 1, given the small sample size, data of the 2 groups (alive, dead) can be shown.
Although conducting mortality univariate analysis yield the same results as comparing alive and dead groups, the results were added to table 1.
- Table 1, tumor number and size should be shown.
This has been added to Table 1.
- Table 1 and 2, treatment response and SAE should be included.
We thank reviewer for this point, However treatment response and SAE ares not added to Table 1 and 2 as these tables only include pre-SIRT factors and the aim of this study was to see how the pre-SIRT factors associate with the mortality after SIRT.
However, we ran the analysis and association between mortality and neither complete (HR 0.90, 95% CI 0.39-2.12, P=0.81) nor partial response (HR 0.88, 95% CI 0.51-1.51, P=0.64) was significant.
- Table 1 and 2, since VAT radiodensity correlates with Child score and not with BCLC, it would be important to see how the its predictive value changes if Child score and vascular invasion and included in the model instead of BCLC.
VAT radiodensity was significantly associated with mortality (HR 2.12, 95% CI, 1.24-3.58, P=0.005) after adjusting for Child-Pugh score and vascular invasion rather than BCLC.
- Definition of vascular invasion should be provided (segmental? Main branch?).
We thank reviewer for this point and added to the results that “Radiological evidence of vascular invasion was present in 19% patients of whom 37% had the involvement of the main portal vein and the remaining 63% experienced invasion of the right or left branches of the portal veins (Lines 115-117).
- Table 2, why the univariate analysis values for HCV changed from Table 2?
We thank the reviewer for this point. In Table 1, all the etiologies were compared to the alcohol-induced cirrhosis; however, in order to maintain the consistency between Tables 1 and 2, we have modified Table 1.
- Figure 1, censored should be shown
We thank reviewer for this point and censored are now shown in Figure 1.
- Table 3, was the ascites present at the time of the SIRT or during the follow-up ?
Those features are as part of pre-SIRT assessment.
- Discussion, the long enrollment time (7 years) should be discussed as a limitation since could affect the way how SIRT or VAT radiodensity were performed or assessed.
We agree with your suggestion, and now we have included in the discussion that:
“Considering the enrollment time and the impact of learning curves on improved treatment delivery, the long inclusion time and subsequently changes in SIRT procedure might be another limitation of this study” (Lines 332-334). The assessment of VAT radiodensity was conducted using CT images taken at L3 and therefore, enrollment time cannot affect the results.
Minor comments
Table 2, specify in the characteristic the cut off for high VAT (not in the legend).
We accept your suggestion and it was modified.
Lines 427- 442, paragraph difficult to read.
Paragraph was modified accordingly.

Reviewer 3 Report
Brief Summary: This manuscript aims to identify the prognostic significance of pre-SIRT body composition features (visceral adipose tissue, subcutaneous adipose tissue and skeletal muscle) on mortality, treatment response and adverse events post-treatment.VAT radiodensity was the only factor that was found to correlate and further analysis shows it is not only correlated with mortality but also development of SAEs. This suggests VAT radiodensity may be used as a surrogate marker to identify which patients may benefit most from SIRT. Overall, I think this is novel study with important prognostic implications. The manuscript is well-written, concise and I believe will be of interest to the cancer community.
The strengths of this paper include variety of objective measures evaluated, SIRT as first therapy.The weaknesses of this paper include limited sample size, single institution and heterogeneity in prior therapies.
Introduction: Lines 64-66: the authors refer to “body composition abnormalities have recently attracted considerable attention” – unclear what is meant by “body composition abnormalities i.e. variations in fat/muscle content, would this be considered an abnormality or an anatomical variation? Also, are there any specific references that can be added to support this statement? Results: Table 2 – These values seem similar to Table 1, Does this table represent the associations of only patients with high VAT radiodensity? If so, should clarify in the table heading. Lines 227-229 – authors report VATI (27 vs 64) and SATI (44 vs 65), are these the comparisons for high vs low radiodensity? If so, this should be clarified. Clinical characteristics of more advanced liver disease –e. features of portal hypertension – were suggestive of higher VAT radiodensity, so in a sense, this is really a surrogate marker for more advanced liver disease. Discussion: The authors do a good job discussing the physiologic association between high VAT radiodensity and poor clinical features/signs of more advanced disease. In lines 295-296 the authors mention that VAT is a sensitive measure for “identifying outcomes of SIRT interventions”, however, there was no association between VAT radiodensity and treatment response, it was only correlated with adverse events - perhaps this can be reworded to clarify. VAT radiodensity appears to the a valuable objective marker, can you please comment on: How reproducible are the results of VAT radiodensity and could it be easily translatable across institutions? How easily can this measurement be obtained? i.e. is it time-consuming for radiologists? Materials and Methods: In the methods lines 395-407, the authors discuss how the body composition variables were defined (VATI, SATI, and SMI), but it is not very clear how VAT radiodensity and mass were determined, i.e.: Was it an Average or median of the HU for VAT radiodensity? Also, there was overlap in the HU range for VAT and subcutaneous adipose tissue – how was this distinguished for measurement purposes?Author Response
Introduction: Lines 64-66: the authors refer to “body composition abnormalities have recently attracted considerable attention” – unclear what is meant by “body composition abnormalities i.e. variations in fat/muscle content, would this be considered an abnormality or an anatomical variation? Also, are there any specific references that can be added to support this statement?
Thank you for this comment. We have modified the sentence as follows and a reference was added:
“Among pre-treatment risk factors that result in poor prognosis, abnormal body composition features such as sarcopenia, and high visceral adiposity have recently attracted considerable attention [5] with a view to provide an objective assessment of the patient’s nutritional and metabolic status.” (Lines 63-66).
Results: Table 2 – These values seem similar to Table 1, Does this table represent the associations of only patients with high VAT radiodensity? If so, should clarify in the table heading.
This table includes variables significant in Table 1 and dichotomous VAT radiodensity. We accept your suggestion and modified the title to “Features associated with mortality by univariate and multivariate cox proportional hazards analysis”.
Lines 227-229 – authors report VATI (27 vs 64) and SATI (44 vs 65), are these the comparisons for high vs low radiodensity? If so, this should be clarified.
We accept your suggestion and modified the sentence as follows:
Regardless of having higher VAT radiodensity, VATI (27±16 vs. 64±29 cm2/m2; P<0.001) and SATI (44±26 vs. 65±37 cm2/m2; P=0.002) were lower in patients with high VAT radiodensity compared to those with low VAT radiodensity.
Clinical characteristics of more advanced liver disease –e. features of portal hypertension – were suggestive of higher VAT radiodensity, so in a sense, this is really a surrogate marker for more advanced liver disease.
We agree that higher VAT radiodensity could be a surrogate marker of more significant advance disease; however, we consider VAT radiodensity is more objective and accurate method to predict outcomes, as its performance was better than conventional scores, such as Child-Pugh. We have mentioned this in the discussion (Lines 263-266).
Moreover, VAT radiodensity was significantly associated with mortality (HR 2.12, 95% CI, 1.24-3.58, P=0.005) after adjusting for Child-Pugh score and vascular invasion.
Discussion: The authors do a good job discussing the physiologic association between high VAT radiodensity and poor clinical features/signs of more advanced disease. In lines 295-296 the authors mention that VAT is a sensitive measure for “identifying outcomes of SIRT interventions”, however, there was no association between VAT radiodensity and treatment response, it was only correlated with adverse events - perhaps this can be reworded to clarify.
We thank the reviewer for this comment and the sentence was reworded as follows:
“This imaging technique offers valuable clinical insights into prognosis of patients with HCC and outlines the necessity to conduct sensitive measures of VAT radiodensity for identifying mortality risk and severe adverse events in patients treated with SIRT in order to target the particular populations that would benefit the most of SIRT” (Lines 274-278).
VAT radiodensity appears to the a valuable objective marker, can you please comment on: How reproducible are the results of VAT radiodensity and could it be easily translatable across institutions? How easily can this measurement be obtained? i.e. is it time-consuming for radiologists?
Thank you for this observation. Using non-contrast abdominal CT images, high intraobserver and interobserver reproducibility has been observed in measuring body composition parameters in clinical populations (Paris MT, Furberg HF, Petruzella S, et al. Influence of Contrast Administration on Computed Tomography-Based Analysis of Visceral Adipose and Skeletal Muscle Tissue in Clear Cell Renal Cell Carcinoma. JPEN Journal of parenteral and enteral nutrition 2018;42:1148-1155) (Lines 396-398).
Using software such as Slice-O-Matic, tissues are selected semi-automatically and the average time to quantify body composition parameters is around eight minutes which can be easily done by any trained observers with the knowledge of anatomy (Cruz RJ Jr, Dew MA, Myaskovsky L, Goodpaster B, Fox K, Fontes P, DiMartini A. Objective radiologic assessment of body composition in patients with end-stage liver disease: going beyond the BMI. Transplantation. 2013 Feb 27;95(4):617-22) (Lines 398-401).
Materials and Methods: In the methods lines 395-407, the authors discuss how the body composition variables were defined (VATI, SATI, and SMI), but it is not very clear how VAT radiodensity and mass were determined, i.e.: Was it an Average or median of the HU for VAT radiodensity? Also, there was overlap in the HU range for VAT and subcutaneous adipose tissue – how was this distinguished for measurement purposes?
Thank you for this suggestion, we now include in the methods that:
“For radiodensity assessment, mean tissue attenuation (HU) was reported for the entire cross sectional area at L3, separately for muscle, SAT and VAT” (Lines 407-409).
Therefore, as SAT and VAT are two separate areas, no overlap exist in reported radiodensity for adipose tissue depots.
